# STRIPAK, a Key Regulator of Fungal Development, Operates as a Multifunctional Signaling Hub

**DOI:** 10.3390/jof7060443

**Published:** 2021-06-01

**Authors:** Ulrich Kück, Valentina Stein

**Affiliations:** Allgemeine und Molekulare Botanik, Faculty for Biology and Biotechnology, Ruhr-University, 44780 Bochum, Germany; valentina.stein@rub.de

**Keywords:** STRIPAK complex, mitophagy, multifunctional signaling hub, fungal development, *Sordaria macrospora*

## Abstract

The striatin-interacting phosphatases and kinases (STRIPAK) multi subunit complex is a highly conserved signaling complex that controls diverse developmental processes in higher and lower eukaryotes. In this perspective article, we summarize how STRIPAK controls diverse developmental processes in euascomycetes, such as fruiting body formation, cell fusion, sexual and vegetative development, pathogenicity, symbiosis, as well as secondary metabolism. Recent structural investigations revealed information about the assembly and stoichiometry of the complex enabling it to act as a signaling hub. Multiple organellar targeting of STRIPAK subunits suggests how this complex connects several signaling transduction pathways involved in diverse cellular developmental processes. Furthermore, recent phosphoproteomic analysis shows that STRIPAK controls the dephosphorylation of subunits from several signaling complexes. We also refer to recent findings in yeast, where the STRIPAK homologue connects conserved signaling pathways, and based on this we suggest how so far non-characterized proteins may functions as receptors connecting mitophagy with the STRIPAK signaling complex. Such lines of investigation should contribute to the overall mechanistic understanding of how STRIPAK controls development in euascomycetes and beyond.

## 1. Introduction

Posttranslational phosphorylation of eukaryotic proteins plays an essential role in modulating their function and is tightly regulated in time and space by the fine-tuned balance between protein kinases and phosphatases [1]. The serine/threonine protein phosphatase PP2A is a heterotrimeric holoenzyme comprising a structural (PP2AA), a catalytic (PP2Ac), and a regulatory B subunit. So far, four different regulatory subunits are known, B, B′, B″, and B‴. Subunit B‴ is designated as striatin and is the name-giving subunit of the striatin-interacting phosphatases and kinases (STRIPAK) signaling complex. This multimeric complex, first characterized in mammalian cells [2], and found to be involved in the regulation of various target proteins, contains a number of other core components. Mass spectrometry analysis primarily characterized different subunits of STRIPAK in mammalian or fungal systems, and various reviews summarize the basic components of the complex [3,4,5,6]. In addition to the PP2A holoenzyme, these are the striatin-interacting proteins STRIP1/2, the mammalian Mps one binder homolog Mob3/phocein, the sarcolemmal membrane-associated protein (SLMAP), and the coiled-coil protein suppressor of IκB kinase-ε (SIKE). These subunits have been characterized in various eukaryotic STRIPAK complexes (for review see [4,5]), and the synonymous designations in euascomycetes are found in Table 1.

Here, we briefly summarize our current knowledge about the function of the STRIPAK complex of ascomycetes, with an emphasis on the role of euascomycetous development. We further refer to recent structural investigations and cellular analyses, which help us understand how STRIPAK acts as a scaffolding complex that cross-links several signal transduction pathways. Finally, current molecular cellular investigations of the STRIPAK homologue in yeast provide insights into a mechanistical hypothesis of how STRIPAK interacts with other signaling pathways at the cellular level, and thus integrates various developmental programs.

## 2. The Structure of STRIPAK Suggests the Integration of Diverse Cellular Signals

Striatin is the central subunit of STRIPAK, and the primary structure of this B‴ subunit is conserved in all eukaryotic systems studied so far [3,4,5,6]. Striatins have four domains: an N-terminal caveolin-binding domain, a coiled-coil domain, a calmodulin-binding domain, and a C-terminal WD40 repeat domain. Important for the cellular localization and complex assembly is the coiled-coil domain, which further mediates its homo- or hetero-oligomerization. Crystal structures of the coiled-coil domain indicated a parallel dimeric but asymmetric conformation of striatin containing a large bend [7].

Although the association of STRIPAK into a heterotrimeric holoenzyme is validated, the overall structure of the complex and knowledge about the stoichiometric composition of the subunits are still cryptic. Assembly of the fungal STRIPAK complex was recently investigated in the euascomycete *Aspergillus nidulans* [8]. Using wild type and STRIPAK subunit deletion strains, these studies showed that the STRIPAK complex is formed from three subcomplexes (SIKE-SLMAP, MOB3, and STRIP1/2-PP2Ac1-PP2AA), where striatin acts as a scaffold at the nuclear envelope. Previous crystallographic analysis had explored the subunit stoichiometry from mammalian STRIPAK, thus promoting our understanding of the different biological functions [9,10]. A further refinement of the structure of a purified human protein complex was very recently determined by cryo-EM, at 3.2-Å resolution [11]. STRIPAK contains four copies of striatin and one copy of each of the other subunits. According to this study, the four WD40 repeat domains from the four striatins interact with different activators or suppressors of other signaling pathways. Included is the Hippo signaling pathway, which is conserved within eukaryotes [12]. The high conservation of STRIPAK subunits allows us to speculate that the structural implications derived from analysis of the mammalian complex can be transferred to STRIPAK of fungal organisms.

## 3. STRIPAK Subunits Have an Impact on Fungal Development

Distinct genes encoding subunits of the STRIPAK complex have been analyzed in a variety of filamentous ascomycetes. Mutations in the corresponding genes lead to severe developmental defects as summarized in Table 1. Here, we summarize how defects in subunits of the STRIPAK complex affect fungal development. Initially, genes for STRIPAK subunits were detected upon investigating the formation of fruiting bodies or hyphal cell fusion (for review see [4]). In *Sordaria macrospora*, a self-fertile homothallic member of the Sordariaceae family, sterile strains were screened for mutant genes, many of which were found to encode STRIPAK subunits. The mutants only generated small immature fruiting bodies, called protoperithecia of about 50 µm in diameter, and contained no ascospores nor asci. Fertility was clearly restored when the wild type subunit gene was introduced into the mutant genome [13]. Such STRIPAK control of fruiting body formation has been observed in a variety of ascomycetes, as summarized in Table 1.

In an analogous mutant screen, *Neurospora crassa* strains with a defect in cell–cell fusion were investigated and the genes responsible were abbreviated as HAM for hyphal anastomosis. Of these, several showed a mutation in genes for STRIPAK subunits, such as HAM-2, -3, and -4 [14,15,16]. Cell–cell fusions are not only the starting process for sexual development and fruiting body formation but seems also to be important in some symbiotic interactions with other organisms, as well as in infection processes by pathogenic species. A common feature of many STRIPAK mutants is a block in the sexual life cycle. However, in heterothallic species, such as *N. crassa*, female but not male fertility is affected by mutant genes for STRIPAK subunits. Furthermore, it has been hypothesized that the lack of hyphal fusions leads to impaired nuclear division during meiosis and to the formation of abnormal ascospores. These observations led to the hypothesis that subunits of STRIPAK are involved in regulating the cell cycle [17]. In this context, it is relevant that not only septation of vegetative hyphae but also of female gametangia (ascocogonia) is apparently controlled by subunits of STRIPAK [18,19,20].

A remarkable effect was observed in *A. nidulans* when proper expression of the velvet complex [21] was investigated in mutants lacking STRIPAK subunits. The mutants showed not only an absence of light-dependent fungal development and secondary metabolite production, but also a reduced stress response [8].

Plant–fungal interactions are another developmental process that is controlled by STRIPAK. *Epichloë festucae* forms a mutualistic symbiotic relationship with the grass *Lolium perenne.* The fungus produces several secondary metabolites thus protecting the host against herbivores. This interaction arises by fungal colonization of intercellular spaces and the leaf surface. Loss of *mobC*, which is homologous to *mob3*, completely abolishes the symbiotic interaction between the fungus and the grass. Instead, *E. festucae* shows extensive hyphal growth within the intercellular tissue of the grass, and in addition, hyphae are found within the vascular bundle tissue, which was never observed with the wild type. Infection with the *mobC* mutant leads to underdeveloped grasses, lacking a functional association with the fungal symbiont [22].

For a long time, plant pathogenic fungi were the subjects of genetic analysis in order to understand their pathogenic mechanisms. These investigations revealed that STRIPAK controls the virulence process in plant pathogens such as *Colletotrichum graminicola*, *Fusarium verticillioides*, *Fusarium virguliforme*, and *Magnaporthe oryzae*. In the corn pathogen *F. verticillioides*, the N-terminal domains of the striatin homologue Fsr1, as well as the STRIP1/2 homologue FvStp1 are important in maize stalk rot virulence [23,24,25]. Similarly, in another corn pathogen, *C. graminicola*, striatin null mutants have functional appressoria; however, the colonization leading to infection is diminished [26]. Virulence was similarly investigated in the soybean pathogen *F. virguliforme.* Here, the striatin homologue FvStr1 controls the colonization of the vascular system, although phloem and xylem vessels of the roots are still infected by mutants [27]. In the rice blast fungus *M. oryzae*, strains lacking the catalytic subunit of PP2A (MoPPG1) were unable to form appressoria for penetrating the host plant [28]. From these examples, it becomes evident that STRIPAK controls in some instances plant–fungal relationships by controlling cell–cell interactions and the formation of infection structures.

So far however, reports are lacking that fungal–animal interactions are controlled by STRIPAK. For example, nematode trapping fungi were recently investigated genetically and shown to be dependent on conserved signaling networks for developing specialized trap structures to capture, kill, and consume nematodes [29,30,31]. We predict that loss of STRIPAK components will result in defects of animal trapping. Similarly, the virulence of approximately 625 fungal species that infect vertebrates [32] were not reported to be controlled by STRIPAK. It may be envisioned that future molecular genetic studies in well-studied human pathogens such as *Aspergillus fumigatus*, *Candida albicans*, *Cryptococcus neoformans*, or *Histoplasma capsulatum* will discover virulence mutants with a defect in genes for subunits of the STRIPAK complex. This assumption is supported by the fact that genes for STRIPAK subunits are very similar between pathogenic and non-pathogenic ascomycetes and basidiomycetes. At least, so far characterized protein domains in homologues of striatin, SLMAP and STRIP1/2 seem to be highly conserved (Appendix A). An exception seems to be the yeast *C. albicans*. This pathogen has a truncated gene for a striatin-like protein, lacking WD40 repeats. With this, it resembles the FAR11 protein from baker’s yeast, which is referred to in the final section [33].

## 4. Dual Localization of STRIPAK Connects Signaling Pathways

The previous section reviewed the diverse developmental processes controlled by STRIPAK. Thus, the question arises as to how this protein complex regulates mechanistically diverse processes within a single organism.

Eukaryotic cells are compartmentalized within membrane-bound organelles, and multiple targeting of proteins or protein complexes can arise through distinct mechanisms during evolution or by stepwise accumulation of mutations within new subunits of a macromolecular complex [34,35]. In various euascomycetes, localization studies have detected STRIPAK subunits at different membranes within the fungal cell, such as the nuclear membrane, the endoplasmatic reticulum (ER), or the mitochondrial outer membranes [8,15,25,36], (Table 2). For *A. nidulans* and *F. verticillium*, more than one cellular localization was reported, and removing striatin by mutation resulted in dislocalization of the remaining STRIPAK subunits [8,25]. Structured illumination microscopy (SIM) in *S. macrospora* showed in detail that PRO45, the homologue of SLMAP, resides in the nuclear envelope, the mitochondrial membrane and the spindle pole body. These results led to the hypothesis that this SLMAP homologue functions as a membrane organizer to mediate signaling by bridging two or more organelles [36]. These findings are compatible to studies with mammalian cells where the SLMAP subunit of STRIPAK resides in the sarcolemma, transverse (T)-tubules, and sarcoplasmic reticulum (SR) of muscle cells, as well as in the outer nuclear envelope, ER, mitochondria, and centrosomes of non-muscle cells [37,38,39,40,41]. Such close associations, for example, between mitochondria and the ER, is believed to transmit calcium signals necessary for balanced cell metabolism [42].

Genetic and molecular analysis in fungi has shown previously that STRIPAK is associated with signaling transduction pathways. In *N. crassa*, for example, subunits of STRIPAK components are involved in the nuclear localization of MAK-1, the downstream kinase of the cell wall integrity (CWI) pathway [15], which supports the hypothesis of crosstalk between the STRIPAK complex and other signaling transduction pathways in fungi. This seems to be consistent with a recent phosphoproteomic analysis, demonstrating that subunits of diverse signaling complexes, such as the target of rapamycin complex 2 (TORC2), nicotinamide adenine dinucleotide phosphate oxidase (NOX), septation initiation network (SIN), CWI and pheromone response (PR) pathway, were phosphorylated or dephosphorylated in a STRIPAK-dependent manner [43,44].

A recent report from the protoascomycete *Saccharomyces cerevisiae* broadens our mechanistic view of how multiple cellular localizations of STRIPAK subunits connect to signaling transduction pathways [45]. In yeast, the factor arrest (FAR) complex is the STRIPAK homologue, which, for example, controls vacuolar protein sorting and pheromone-induced cell cycle arrest. Subpopulations of FAR reside either in the ER or in the mitochondria, and this localization is determined by the tail-anchor domain of Far9/10, the homologue of SLMAP. When located in the ER, FAR regulates the TORC2 signaling pathway, while its location in the mitochondrial membrane inhibits mitophagy, a process were mitochondria are selectively degraded by autophagy. The yeast mitophagy receptor Atg32p was shown to interact with Far8p, the homologue of striatin. Atg32p is phosphorylated by casein kinase 2 (CK2), which is essential for mitophagy. The phosphorylation is counteracted by the FAR complex, causing to dissociate from Atg32p upon mitophagy stimuli [46].

So far, a homologue of Atg32p has neither been detected in mammals nor in filamentous euascomycetes [47,48]. However, for mammalian cells, eight mitophagy receptors were described, of which one, Bcl2-L-13, is functionally homologous to Atg32p [49,50]. Like Atg32p, Bcl2-L-13 carries a C-terminal transmembrane domain (TM), an amino acid acidic cluster, and an LC3-interacting region (LIR), which binds ATG8 and corresponds to the Atg8-interacting motifs (AIM) in yeast [51]. Finally, Atg32p and Bcl2-L-13 are characterized by a CK2 phosphorylation site and a PP2A docking site. Atg32p additionally carries an Atg11 binding region (A11BR), which is predicted to bind Atg11p.

In diverse genomes of euascomycetes, no homologue of the Atg32p or Bcl2-L-13 gene has been detected [48]. Therefore, we searched for *S. macrospora* genes encoding proteins with sequence motifs that are characteristically found in Atg32p and/or Bcl2-L-13. Indeed, we detected 27 genes (Appendix A) in the complete and well-curated *S. macrospora* genome encoding proteins with these domains or motifs [52,53]. Among these, six genes were found exclusively in euascomycetes (Appendix A). Interestingly, one of the candidate proteins (SMAC_04227) is present in the recently characterized library of STRIPAK-dependent phosphorylated proteins from *S. macrospora* [43,44]. Thus, we hypothesize that the so far uncharacterized *S. macrospora* protein SMAC_04227 is potentially functioning as a mitophagy receptor, which similar to Atg32p from yeast, connects the STRIPAK complex to mitophagy. Within euascomycetes, SMAC_04227 shows a high sequence similarity to other homologues (Appendix A), and in future functional investigations it will be worth testing whether the phosphorylation status of SMAC_04227 affects an interaction with STRIPAK subunits. Functionally similar to yeast, the phosphorylation of SMAC_04227 or its homologues may promote mitophagy, a process which is being intensively investigated in euascomycetes [47,48,54,55,56,57].

We predict that future investigations will address the question, how STRIPAK controls pathogenic and symbiotic interaction of fungi with plants, animals, including humans, and other microorganisms. Finally, we do not know what regulates the different cellular locations of STRIPAK. This question is associated with the problem of how phosphorylation and dephosphorylation controls the interaction or dissociation of STRIPAK with other signaling complexes. We anticipate that research in this direction will benefit from advances in mass spectrometry-based methods for more comprehensive site-specific phosphorylation profiling. This was recently demonstrated when absolute quantification by parallel-reaction monitoring (PRM) was applied to analyze phosphorylation site occupancy in signaling components of the SIN signaling pathway [58]. In conclusion, well established experimental systems in diverse euascomycetes should contribute to answering these questions and will contribute to the overall mechanistic understanding of how STRIPAK regulates not just euascomycetes development, but in eukaryotes in general.

**Table 1 jof-07-00443-t001:** List of genes encoding STRIPAK subunits and STRIPAK associated kinases, which are all involved in different developmental processes in euascomycetes.

STRIPAK Subunit	Gene Designation	Developmental Process										Organism	Reference
		Ascospore Formation ^1^	Cell Fusion	Conidia Formation	Essential ^2^	Fruiting Body Formation	Patho- Genicity	Primary and Secondary Metabolism	Septation ^3^	Symbiotic Interaction	Vegetative Growth ^4^		
**Mob3/Phocein**	*ChMOB3*										X	*C.h.*	[59]
	*mob-3*		X	X		X						*N.c.*	[60,61]
	*mobC*		X	X						X	X	*E.f.*	[22]
	*sipA*	X		X				X			X	*A.n.*	[8]
	*Smmob3*		X			X					X	*S.m.*	[62]
**PP2AA**	*pp2a-a*				X							*N.c.*	[15]
	*pp2aa*				X							*S.m.*	[20]
	*sipF*				X							*A.n.*	[8]
**PP2Acα/β**	*CPP1*	X		X				X			X	*F.ve.*	[25,63]
	*Moppg1*			X			X				X	*M.o.*	[28]
	*pp2a*			X				X			X	*T.r.*	[64]
	*pp2Ac1*		X			X			X		X	*S.m.*	[20]
	*ppg-1*		X	X		X					X	*N.c.*	[15]
	*sipF*			X		X		X			X	*A.n.*	[8]
**SIKE**	*sci1*		X			X					X	*S.m.*	[65]
	*sipB*			X		X		X			X	*A.n.*	[8]
**SLMAP**	*ham-4*	X	X									*N.c.*	[17]
	*PaPro45*		X			X					X	*P.a.*	[66]
	*pro45*		X			X					X	*S.m.*	[36]
	*sipD*			X		X		X			X	*A.n.*	[8]
**Striatin**	*FgFSR1*					X	X					*F.g.*	[23]
	*FSR1*					X	X				X	*F.ve.*	[23,25]
	*FvSTR1*			X		X	X					*F.vi.*	[27]
	*ham-3*	X	X			X						*N.c.*	[15,17]
	*pro11*		X			X			X			*S.m.*	[67]
	*strA*			X		X		X			X	*A.n.*	[8]
	*str1*		X	X		X	X				X	*C.g.*	[26]
**STRIP1/2**	*ham-2*	X	X			X						*N.c.*	[17]
	*PaPro22*		X			X					X	*P.a.*	[66]
	*pro22*		X			X			X			*S.m.*	[18]
	*sipC*			X		X		X			X	*A.n.*	[8]
	*FvSTP1*					X	X				X	*F.ve.*	[25]
**STRIPAK associated kinases**													
**GCKIII**	*Fg07344*	X		X			X				X	*F.g.*	[68]
	*sepL*			X					X		X	*A.n.*	[69]
	*sid-1*, *mst-1*					X			X			*N.c.*	[70]
	*Smkin3*, *Smkin24*		X			X			X		X	*S.m.*	[19,71]

^1^ Ascospore formation may be lacking or abnormal ascospore are generated. ^2^ Deletion of the genes is lethal. ^3^ includes hyphal or ascogonal septation. ^4^ includes reduced growth rates or abnormal hyphal morphology. “x” indicates processes, in which the protein is involved. “Blank cells” indicate that the proteins have not been investigated or are not relevant for the corresponding processes. Abbreviations: *A.n.*, *Aspergillus nidulans*; *C.g.*, *Colletotrichum graminicola*; *C.h.*, *Colletotrichum higginsianum*; *E.f.*, *Epichloë festucae*; *F.g.*, *Fusarium graminearum*; *F.ve.*, *Fusarium verticillioides*; *F.vi.*, *Fusarium virguliform*; *M.o.*, *Magnaporthe oryzae*; *N.c.*, *Neurospora crassa*; *P.a.*, *Podospora anserina*; *S.m.*, *Sordaria macrospora*; *T.r.*, *Trichoderma reesei*.

**Table 2 jof-07-00443-t002:** Localization of STRIPAK subunits in various euascomycetes.

Subunit		Organism	Localization	Reference
**Mob3/Phocein**	MOB-3	*N. crassa*	Nuclear envelope	[15]
	SmMOB3	*S. macrospora*	Nuclear envelope	[72]
	SipA	*A. nidulans*	ER (string-like extensions), nuclear envelope (dependant on StrA), nucelus	[8]
**PP2Acα/β**	PPG-1	*N. crassa*	Nucleus	[15]
	SipE	*A. nidulans*	Nuclear envelope (dependant on StrA)	[8]
**SIKE**	SCI1	*S. macrospora*	Nuclear envelope	[65]
	SipB	*A. nidulans*	Nuclear envelope (dependant on StrA)	[8]
**SLMAP**	HAM-4	*N. crassa*	Nuclear envelope	[15]
	PRO45	*S. macrospora*	Nuclear envelope, the mitochondrial membrane and the spindle pole body	[36]
	SipD	*A. nidulans*	Nuclear envelope (dependant on StrA)	[8]
**Striatin**	Fsr1	*F. verticillioides*	ER, nuclear envelope, vacuolar membranes or late endosomes in	[25]
	HAM-3	*N. crassa*	Nuclear envelope	[15]
	PRO11	*S. macrospora*	Nuclear envelope	[65]
	StrA	*A. nidulans*	ER, nuclear envelope,	[8,73]
**STRIP1/2**	HAM-2	*N. crassa*	Nuclear envelope	[15]
	PRO22	*S. macrospora*	dynamic tubular and vesicular vacuolar network	[18]
	SipC	*A. nidulans*	Nuclear envelope (dependant on StrA)	[8]
**GCKIII**	SID-1	*N. crassa*	Spindle pole body (SPB), septa	[70]
	SmKIN3	*S. macrospora*	Spindle pole body (SPB), septa	[58]

## Data Availability

Not applicable.

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
