# Peer review of "STRIPAK, a Key Regulator of Fungal Development, Operates as a Multifunctional Signaling Hub"

_jof, 2021, doi:10.3390/jof7060443_

Round 1
Reviewer 1 Report
In their perspective article, the authors provide a concise and highly informative overview of the structure and function of the conserved STRIPAK signaling complex. The manuscript focuses on fungal STRIPAK but provides sufficient references to other organisms, thereby highlighting the conservation of this signaling hub. This review provides an extremely useful overview suited for a broad audience interested in this emerging field.
The manuscript has been carefully prepared and is very well structured. The following suggestions and questions aim mostly at clarifying some of the statements and should be easily addressed by rephrasing some of the sections:
In the introduction, the sentence "The serine/threonine protein phosphatase PP2A forms a heterotrimeric holoenzyme with a structural (PP2AA), a catalytic (PP2Ac), and a regulatory B subunit" is a little confusing, since it sounds as if PP2A is one protein, which associates with three other factors, which would be inconsistent with a heterotrimeric complex. How about ""The serine/threonine protein phosphatase PP2A is a heterotrimeric holoenzyme comprising a structural (PP2AA), a catalytic (PP2Ac), and a regulatory B subunit"?
In Section 2, the authors state that "Striatin (...) is conserved in all organismal systems studied so far". Does this also include non-eukaryotic organisms?
In the following parts of the same section, I got, unfortunately, pretty lost in the summary of the Aspergillus data. Especially the statement "the heptameric complex is formed from three subcomplexes" left me pretty confused. Even with the subsequent statements it did not become clear to me, what the seven parts of "heptameric" and what the three subcomplexes are. When I revisited the original publication, I noticed that the structure was described as a hexameric complex of three subunits. This I also found confusing, since the respective cartoon shows a complex of seven proteins, which would indeed be heptameric. The authors might want to edit this section for improved clarity and to solve this heptameric, hexameric conundrum. Maybe describe what each of the three subunits comprises, so that the number seven becomes clear. I personally would find a little cartoon in the supplemental material extremely helpful. For readers not too familiar with the field the whole composition of complexes and subcomplexes can become a little confusing and this article could be a fantastic opportunity to provide some assistance in understanding of these issues.
In Section 3 the authors state: "Cell-cell fusions are not only the starting process for sexual development and fruiting body formation, but also the symbiotic interaction with other organisms, as well as the infection process by pathogenic species". To me this sounds as if the deficiencies in cell-cell fusion are the cause for the defects in pathogenicity or symbiosis. When revisiting some of the original publications, I got the impression that merely a correlation between these defects exists, but that the role of STRIPAK in plant-fungus interactions could also be fully independent of the fusion defect. The authors might consider to rephrase this section slightly to tone their statement somewhat down.
In the last paragraph of section 3, the authors report studies on nematode trapping fungi. However, it is not clearly stated what the studies found concerning the role of STRIPAK. This left me a little confused. Did they show that STRIPAK has no function in animal trapping?
Table 1 took me a while to clearly understand what is shown. Maybe add some information to the legend, such as "x indicates processes, in which the protein is involved". For fungi not pathogenic or symbiontic, consider adding "n.a." or "N/A" to the respective cells.
Minor comments:
I found the wording "suggest how" in the abstract slightly confusing. It´s use sounds unusual to me. Maybe double check.
Section 3 line 5: correct "inverstigating"
The very last sentence: "(...) how STRIPAK regulates (...) in eukaryotes in general". The wording seems a little off. Consider rephrasing.
Figure S4: use the term asterisk instead of star
Author Response
Our responses to the reviewers’ comments.
Reviewer #1
In their perspective article, the authors provide a concise and highly informative overview of the structure and function of the conserved STRIPAK signaling complex. The manuscript focuses on fungal STRIPAK but provides sufficient references to other organisms, thereby highlighting the conservation of this signaling hub. This review provides an extremely useful overview suited for a broad audience interested in this emerging field.
We thank the reviewer for his very positive evaluation.
The manuscript has been carefully prepared and is very well structured. The following suggestions and questions aim mostly at clarifying some of the statements and should be easily addressed by rephrasing some of the sections:
In the introduction, the sentence "The serine/threonine protein phosphatase PP2A forms a heterotrimeric holoenzyme with a structural (PP2AA), a catalytic (PP2Ac), and a regulatory B subunit" is a little confusing, since it sounds as if PP2A is one protein, which associates with three other factors, which would be inconsistent with a heterotrimeric complex. How about ""The serine/threonine protein phosphatase PP2A is a heterotrimeric holoenzyme comprising a structural (PP2AA), a catalytic (PP2Ac), and a regulatory B subunit"?
Accepted, changes were done on line 27-29 of the revised version
In Section 2, the authors state that "Striatin (...) is conserved in all organismal systems studied so far". Does this also include non-eukaryotic organisms?
Accepted, rephrasing was done on line 52 of the revised version
In the following parts of the same section, I got, unfortunately, pretty lost in the summary of the Aspergillus data. Especially the statement "the heptameric complex is formed from three subcomplexes" left me pretty confused. Even with the subsequent statements it did not become clear to me, what the seven parts of "heptameric" and what the three subcomplexes are. When I revisited the original publication, I noticed that the structure was described as a hexameric complex of three subunits. This I also found confusing, since the respective cartoon shows a complex of seven proteins, which would indeed be heptameric. The authors might want to edit this section for improved clarity and to solve this heptameric, hexameric conundrum. Maybe describe what each of the three subunits comprises, so that the number seven becomes clear. I personally would find a little cartoon in the supplemental material extremely helpful. For readers not too familiar with the field the whole composition of complexes and subcomplexes can become a little confusing and this article could be a fantastic opportunity to provide some assistance in understanding of these issues.
Rephrasing of this sentence was done on line 62 and 63. We abstain of showing a cartoon, since this was already done in references 5 and 8
In Section 3 the authors state: "Cell-cell fusions are not only the starting process for sexual development and fruiting body formation, but also the symbiotic interaction with other organisms, as well as the infection process by pathogenic species". To me this sounds as if the deficiencies in cell-cell fusion are the cause for the defects in pathogenicity or symbiosis. When revisiting some of the original publications, I got the impression that merely a correlation between these defects exists, but that the role of STRIPAK in plant-fungus interactions could also be fully independent of the fusion defect. The authors might consider to rephrase this section slightly to tone their statement somewhat down.
We toned down our statement by rephrasing on line 91-93 and 149
In the last paragraph of section 3, the authors report studies on nematode trapping fungi. However, it is not clearly stated what the studies found concerning the role of STRIPAK. This left me a little confused. Did they show that STRIPAK has no function in animal trapping?
Rephrasing was done as requested on line 154-155
Table 1 took me a while to clearly understand what is shown. Maybe add some information to the legend, such as "x indicates processes, in which the protein is involved". For fungi not pathogenic or symbiontic, consider adding "n.a." or "N/A" to the respective cells.
We added the following legend ““x” indicates processes, in which the protein is involved. “Blank cells” indicate that the proteins have not been investigated or are not relevant for the corresponding processes.
Minor comments:
I found the wording "suggest how" in the abstract slightly confusing. It´s use sounds unusual to me. Maybe double check.
Phrase was suggested by a native English speaker
Section 3 line 5: correct "inverstigating"
Was done
The very last sentence: "(...) how STRIPAK regulates (...) in eukaryotes in general". The wording seems a little off. Consider rephrasing.
Phrase was suggested by a native English speaker. Therefore we prefer to keep the text as was done.
Figure S4: use the term asterisk instead of star
Was done
Reviewer 2 Report
The manuscript is dedicated to the role of multi subunit complex of the striatin-interacting phosphatases and kinases (STRIPAK). Authors describe in details the structure and the control function of STRIPAK in broad-range of developmental processes in euascomycetes, localization of STRIPAK and the way it can influence signaling pathways.
In general the manuscript is well-written and detailed. But more references are needed in the Introduction section. The statement "Striatin is the central subunit of STRIPAK, and the primary structure of this B’’’ subunit is conserved in all organismal systems studied so far." is not also supported by references in the text. Please, add some references. The conclusion is too short, some extension is required.
Author Response
Our responses to Reviewer #2
The manuscript is dedicated to the role of multi subunit complex of the striatin-interacting phosphatases and kinases (STRIPAK). Authors describe in details the structure and the control function of STRIPAK in broad-range of developmental processes in euascomycetes, localization of STRIPAK and the way it can influence signaling pathways.
We thank the reviewer for his positive judgement
In general the manuscript is well-written and detailed. But more references are needed in the Introduction section. The statement "Striatin is the central subunit of STRIPAK, and the primary structure of this B’’’ subunit is conserved in all organismal systems studied so far." is not also supported by references in the text. Please, add some references.
We have added 4 references to this section
The conclusion is too short, some extension is required.
The final perspective was extended as requested by the reviewer on line 236-238